# Measuring inter-protein pairwise interaction energies from a single native mass spectrum by double-mutant cycle analysis

Miri Sokolovski[1], Jelena Cveticanin[2], Déborah Hayoun[2], Ilia Korobko[1], Michal Sharon[2] & Amnon Horovitz[1]

The strength and specificity of protein complex formation is crucial for most life processes and is determined by interactions between residues in the binding partners. Double-mutant cycle analysis provides a strategy for studying the energetic coupling between amino acids at the interfaces of such complexes. Here we show that these pairwise interaction energies can be determined from a single high-resolution native mass spectrum by measuring the intensities of the complexes formed by the two wild-type proteins, the complex of each wild-type protein with a mutant protein, and the complex of the two mutant proteins. This native mass spectrometry approach, which obviates the need for error-prone measurements of binding constants, can provide information regarding multiple interactions in a single spectrum much like nuclear Overhauser effects (NOEs) in nuclear magnetic resonance. Importantly, our results show that specific inter-protein contacts in solution are maintained in the gas phase.

[1] Department of Structural Biology, Weizmann Institute of Science, Rehovot 761001, Israel. [2] Department of Biomolecular Sciences, Weizmann Institute of Science, Rehovot 761001, Israel. Miri Sokolovski, Jelena Cveticanin, and Déborah Hayoun contributed equally to this work. Correspondence and requests for materials should be addressed to M.S. (email: michal.sharon@weizmann.ac.il) or to A.H. (email: amnon.horovitz@weizmann.ac.il)

Methods such as X-ray crystallography, electron microscopy, and nuclear magnetic resonance can be used to determine the structures of protein complexes at high resolution. Nevertheless, complementary methods are needed for determining the energetics of binding reactions and for obtaining structural information in cases that are not amenable to analysis by these methods owing, e.g., to low solubility, large size, and/or high flexibility. One such complementary method that is widely used is double-mutant cycle (DMC) analysis[1]. In this method, two residues of interest are mutated individually and in combination and the energetic effects of the mutations (e.g., on binding or folding) are measured. If the effect of the double mutation differs from the sum of effects of the corresponding single mutations then an interaction, either direct or indirect, between the two residues is inferred. Under favorable circumstances, the deviation from additivity, termed the coupling energy, can be used to estimate the strength of intramolecular[2, 3] and intermolecular[4] pairwise interactions in proteins or protein–ligand complexes. Coupling energies can also be used as constraints (much like nuclear Overhauser effects (NOEs) in nuclear magnetic resonance) to determine protein structures, e.g., when docking two proteins to each other[5]. Recent applications of the DMC method in molecular recognition studies include combining it with cryo-electron microscopy to identify the binding site of capsaicin in the TRPV1 ion channel[6] and demonstrating an affinity-specificity tradeoff in PDZ–peptide interactions[7]. Here we describe a native mass spectrometry-based approach that simplifies the application and enhances the power of DMC analysis for studying protein interactions. Our results also show that specific inter-residue interactions are maintained in the gas phase.

## Results

**Native mass spectrometry can simplify DMC analysis.** The application of DMCs for detecting and quantifying inter-protein pairwise interactions between residue i in protein X and residue j in protein Y involves creating a cycle that comprises the four possible complexes that can be formed by the two wild-type proteins, $X_i$ and $Y_j$, and the two corresponding single mutants, $X_m$ and $Y_m$, (Fig. 1). The change in the free energy of binding upon the mutation i→m, when leaving residue j unchanged may be expressed relative to the wild-type protein complex as $\Delta G(ij{\to}mj)$. Likewise, the change in the binding free energy upon the mutation i→m, when residue j has already been mutated to m, may be expressed as $\Delta G(im{\to}mm)$. The difference between $\Delta G(ij{\to}mj)$ and $\Delta G(im{\to}mm)$, which is equal to the difference between $\Delta G(ij{\to}im)$ and $\Delta G(mj{\to}mm)$, provides a measure of the coupling energy, $\Delta\Delta G_{int}(ij)$, between residue i in protein X and residue j in protein Y. This coupling energy can be expressed in terms of binding constants as follows:

$$\Delta\Delta G_{int}(ij) = RT\ln\left(K_{ij}K_{mm}/K_{im}K_{mj}\right), \qquad (1)$$

where $K_{ij}$ is the binding constant for the two wild-type proteins, $X_i$ and $Y_j$, $K_{mm}$ is the binding constant for the two corresponding single mutants $X_m$ and $Y_m$ and $K_{im}$ and $K_{mj}$ are the respective binding constants for $X_i$ with $Y_m$ and $X_m$ with $Y_j$.

Identification and quantification of an inter-molecular pairwise interaction using the DMC approach is labor intensive since it requires generating binding isotherms for the four complexes in each cycle using methods such as isothermal calorimetry or surface plasmon resonance. It is also error-prone because unavoidable errors in protein concentrations lead to errors in the estimates of the binding constants and, thus, to a cumulative

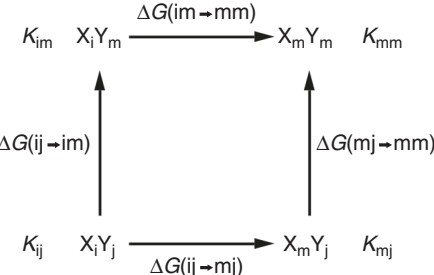

**Fig. 1** Double-mutant cycle for measuring inter-protein pairwise interactions. X and Y stand for two proteins and i and j for two residues in these respective proteins that are mutated to some other residue (e.g., alanine) designated by 'm'. The binding constants corresponding to the formation of the four complexes in the cycle and the free energies changes upon the mutations are also indicated in the scheme

error in the estimate of the value of the coupling energy, $\Delta\Delta G_{int}(ij)$. The native mass spectrometry (MS) approach for DMC analysis that is described here circumvents the need for the determination of the individual binding constants and provides a way to measure $\Delta\Delta G_{int}(ij)$ directly from a single spectrum. It is based on the realization that substitution of each of the binding constants in Eq. 1 by the appropriate ratio of free protein and complex concentrations (e.g., $K_{ij} = [X_iY_j]/[X_i][Y_j]$) leads to the following expression:

$$\Delta\Delta G_{int}(ij) = RT\ln\left([X_iY_j][X_mY_m]/[X_iY_m][X_mY_j]\right), \qquad (2)$$

where the concentrations of all the free species cancel out.

Native MS is a method based on the ability to transfer protein complexes to the gas phase without disrupting them[8–12]. The high resolution afforded by the Orbitrap platform enabled us to determine the intensities of all four complexes simultaneously from a single mass spectrum. Hence, by mixing the two wild-type proteins, $X_i$ and $Y_j$, and the two corresponding single mutants $X_m$ and $Y_m$, and then measuring the intensities of all four possible co-existing complexes using native MS one can obtain a direct estimate of $\Delta\Delta G_{int}(ij)$ that does not depend on knowing the concentrations of the free species ($[X_i]$, $[Y_j]$, $[X_m]$, and $[Y_m]$).

**Targeting inter-protein interactions in the E9:Im2 complex.** The applicability of this native MS-based approach for measuring pairwise coupling energies was tested for the well-characterized interaction of colicin E9 endonuclease with the bacterial immunity protein Im2 with which it forms a 1:1 complex[13]. Two DMCs were constructed: one for the interaction of Asn34 in Im2 with Ser84 in E9 and a second for the interaction of Asp33 in Im2 with Phe86 in E9. Phe86 is a main hot spot residue in the interface of E9 with Im9[13]. In both cycles, all the mutations were to alanine as before[13]. The coupling energies of the two pairwise interactions were determined by the native MS-based approach for DMC analysis described here and, in comparison, by measuring all the required individual binding constants using isothermal titration calorimetry.

**Assignment of the peaks in the native MS spectra.** A prerequisite for DMC analysis by native MS is that the changes in mass upon mutation are sufficiently large so that the masses of the different complexes forming the cycle can be well resolved. In the case of the cycle for the interaction of Asn34 in Im2 with Ser84 in E9, this condition was initially not met. We, therefore, combined the mutation Ser84→Ala in E9 with

**Table 1 Association constants for the interaction of E9 variants with Im2 variants determined by isothermal calorimetry[a]**

| Complex | $K_a$ ($\mu M^{-1}$) | Average values[b] $K_a$ ($\mu M^{-1}$) | Literature values[13] $K_a$ ($\mu M^{-1}$) |
|---|---|---|---|
| E9 wild type–Im2 wild type | 1.08 (±0.09) 1.36 (±0.21) 1.23 (±0.14) | 1.22 (±0.14) | 4.83 (±0.68) |
| E9 wild type–Im2$_{D33A}$ | 8.33 (±0.08) 14.8 (±0.25) 17.0 (±0.21) | 13.4 ± 4.5 | $1.00 \times 10^3$ |
| E9$_{F86A}$–Im2 wild type | 1.68 (±0.03) × $10^{-2}$ 1.16 (±0.02) × $10^{-2}$ 1.22 (±0.02) × $10^{-2}$ | 1.35 (±0.28) × $10^{-2}$ | 6.94 (±0.05) × $10^{-1}$ |
| E9$_{F86A}$–Im2$_{D33A}$ | 1.66 (±0.16) 1.63 (±0.16) 1.55 (±0.17) | 1.61 (±0.06) | 8.33 (±0.15) |
| E9 wild type–Im2$_{N34A}$ | 1.70 (±0.12) 1.30 (±0.12) 1.63 (±0.16) | 1.54 (±0.21) | 4.34 (±0.05) × $10^{-1}$ |
| E9$_{S84A/H131A/R132A}$–Im2 wild type | 2.34 (±0.66) 1.35 (±0.36) 1.37 (±0.49) 1.84 (±0.33) 2.23 (±0.31) | 1.83 (±0.46) | N.D. |
| E9$_{S84A}$–Im2 wild type | 1.29 (±0.14) 1.36 (±0.36) 1.33 (±0.25) 1.65 (±0.43) 1.41 (±0.40) | 1.41 (±0.14) | 7.14 (±0.05) |
| E9$_{S84A}$–Im2$_{N34A}$ | 2.03 (±0.50) 2.53 (±0.90) 2.94 (±0.18) 3.12 (±0.38) 2.11 (±0.32) | 2.55 (±0.49) | N.D. |

[a]A value of ~250 $\mu M^{-1}$ was obtained for the association constant of wild-type E9 to the Im2$_{D33A}$ mutant when using 50 mM MOPS buffer (pH 7.0) containing 200 mM NaCl as before[13]. The discrepancies between the values reported here and the literature values are, therefore, most likely due to the different buffers and salt concentrations used in the two studies.
[b]Values ± standard deviations are reported

two additional mutations, His131→Ala and Arg132→Ala, in order to increase the mass difference. His131 and Arg132 are in the flexible tail of E9, thereby suggesting that they are not involved in any inter- or intra-protein interactions[14]. This was confirmed by showing that the affinity of the E9 Ser84→Ala mutant for Im2 is unchanged by the tail mutations (Table 1 and Supplementary Fig. 1). Introducing such mutations, therefore, provides a general strategy for increasing mass differences when needed. Native MS analysis confirmed that the differences in mass between wild type and mutant E9 or wild type and mutant Im2 can indeed be resolved (Fig. 2). Importantly, the results also show that the wild type and mutant variants of each protein (Fig. 2 and Table 2) and their complexes (Supplementary Tables 1 and 2) have similar ionization efficiencies so that the ratios of their intensities correspond to their concentration ratios in solution.

Given the results in Fig. 2, we proceeded to analyze mixtures containing different ratios of the four wild type and mutant E9 and Im2 proteins that combine to form the different complexes in each of the respective cycles (Figs. 3 and 4). The results show that, although the main peaks corresponding to the respective complexes in each cycle are well resolved from each other, there is some overlap in the minor peaks. The existence of several peaks is most likely due to bound ethylenediammonium ions and not to any variations in the protein sequences or post-translational modifications since only a single peak with a mass corresponding to the expected calculated mass was observed upon MS analysis of each of the denatured proteins (Supplementary Fig. 2). The concentration of each complex,

therefore, had to be determined by summing the areas corresponding to both its major peak and all its minor peaks. Consequently, correct assigning of the peaks in the overlapping regions was necessary in order to obtain reliable estimates of the concentrations. These assignments were achieved by determining how the intensities of the peaks in question change as a function of the mixing ratio. Importantly, inspection of Eq. 1 shows that the values of the coupling energies do not depend on the mixing ratio.

**Coupling energies from native MS and isothermal calorimetry.** Given the relative concentrations of the four complexes that form each of the cycles, it was possible to calculate the coupling energies corresponding to them using Eq. 2 (Tables 3 and 4). The values obtained for the two different charge states and the various concentration ratios were found to be similar as expected. The average values for the interactions of Asn34 in Im2 with Ser84 in E9 (Table 3) and Asp33 in Im2 with Phe86 in E9 (Table 4) are $-0.04 \pm 0.03$ and $1.21 \pm 0.10$ kcal mol$^{-1}$, respectively. These values are in very good agreement with values of $0.06 \pm 0.21$ and $1.43 \pm 0.25$ kcal mol$^{-1}$ that we obtained for these respective interactions using isothermal calorimetry (Fig. 5 and Table 1). The value of about zero measured for the coupling energy of Asn34 in Im2 with Ser84 in E9 is not surprising given the relatively large distance separation between these residues in the complex structure (the closest distance is 8 Å between the OD1 atom of Asn34 and the O atom of Ser84[14]). The positive coupling energy of more than 1 kcal mol$^{-1}$ found

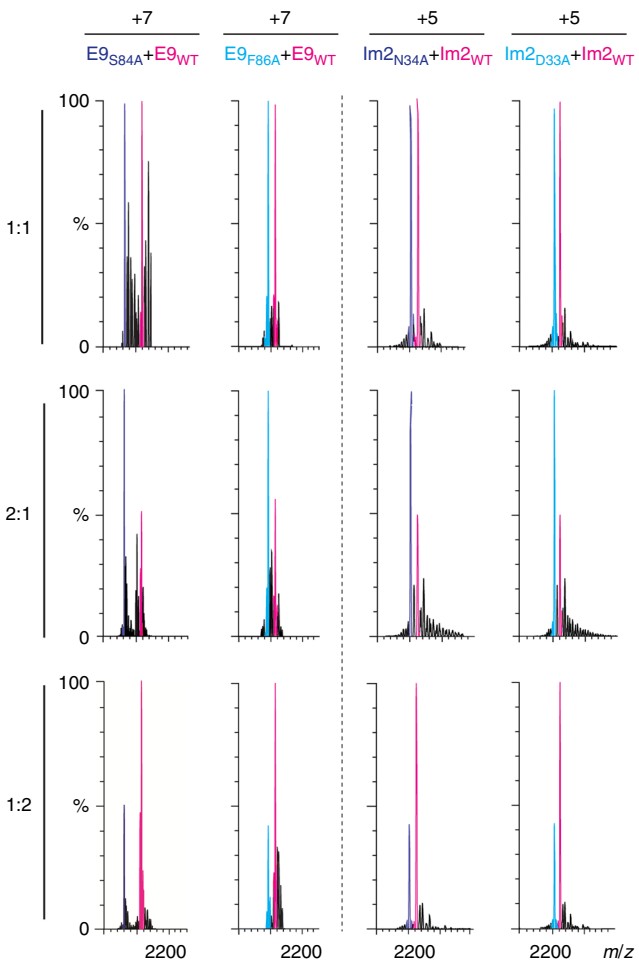

**Fig. 2** Peak areas reflect solution concentrations of proteins. Comparison of the ratios of peak areas of the mutant and wild-type proteins with their relative respective solution concentrations of 1:1, 2:1, and 1:2 shows that they are in excellent agreement. Expansions of charge states +7 and +5 are shown for the E9 and Im2 variants, respectively

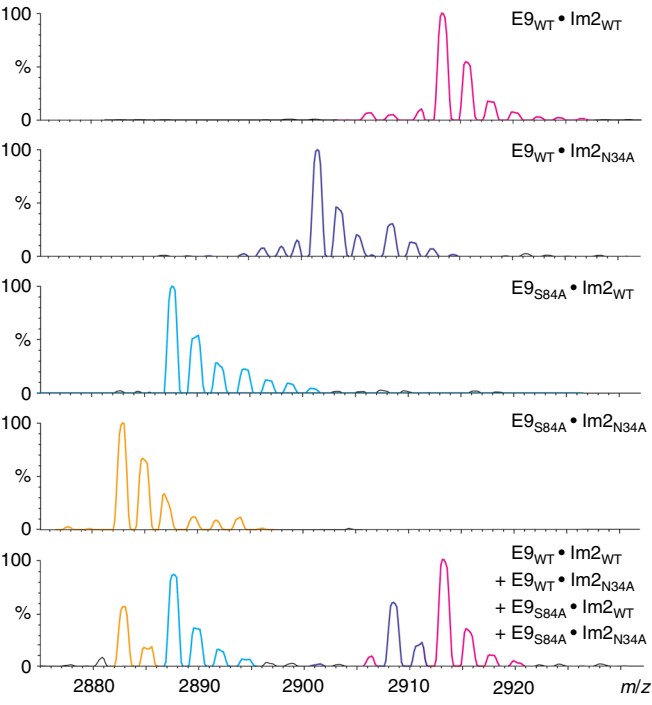

**Fig. 3** Interaction of Ser84 in E9 with Asn34 in Im2 studied by native MS. Representative mass spectra of the +9 charge state are shown for the four pairs of complexes: E9$_{WT}$ • Im2$_{WT}$ (*magenta*), E9$_{WT}$ • Im2$_{N34A}$ (*purple*), E9$_{S84A}$ • Im2$_{WT}$ (*cyan*), and E9$_{S84A}$ • Im2$_{N34A}$ (*yellow*). All these complexes were formed using equal concentrations of the two binding partners. The *spectrum in the bottom panel* corresponds to all four proteins mixed together at relative respective concentrations of E9$_{WT}$, E9$_{S84A}$, Im2$_{WT}$, and Im2$_{N34A}$ of 1:2:2:1, respectively. The highly resolved mass spectrum generated for the mixture of E9$_{WT}$, E9$_{S84A}$, Im2$_{WT}$, and Im2$_{N34A}$ enabled us to distinguish between the four different complexes that are formed. Note that some peaks, which are present when only one complex is formed, are absent when all four complexes are formed together because each protein now forms two different complexes at a ratio that depends on the concentrations of all four proteins and their affinities

| Table 2 Ratios of peak areas reflect the relative concentrations in solution of the measured proteins | | | | |
|---|---|---|---|---|
| **Concentration ratio** | **Peak area ratio** | | | |
| | **E9$_{S84A}$:E9$_{WT}$** | **E9$_{F86A}$:E9$_{WT}$** | **Im2$_{N34A}$:Im2$_{WT}$** | **Im2$_{D33A}$:Im2$_{WT}$** |
| 1:1 | 1.03 ± 0.09 | 0.97 ± 0.03 | 0.98 ± 0.02 | 1.02 ± 0.02 |
| 2:1 | 2.05 ± 0.16 | 2.04 ± 0.07 | 1.97 ± 0.14 | 2.01 ± 0.19 |
| 1:2 | 0.49 ± 0.02 | 0.49 ± 0.04 | 0.51 ± 0.02 | 0.52 ± 0.01 |

The ratios of peak areas of the mutant and wild-type proteins (Fig. 2) are in excellent agreement with their concentration ratios of 1:1, 2:1, and 1:2 in solution

for Asp33 in Im2 and Phe86 in E9 is consistent with a stabilizing and possibly cooperative anion–aromatic interaction between the O and OD2 atoms of Asp33 and the respective Cz and CE1 atoms of Phe86, which are separated in the complex structure by about 4.3 and 5.5 Å, respectively[14]. Cation-pi pairs in proteins have been documented widely but much less attention has been paid to anion–aromatic interactions between negatively charged side chains and the positive electrostatic potential associated with the ring edge of aromatic groups[15, 16]. The negative coupling energy of about −1.7 kcal mol$^{-1}$ reported for the interaction of Asp33 in Im2 and Phe86 in E9 in previous work[13]

suggests that the different buffer conditions employed in the two studies may have caused conformational switching between a favorable (negatively charged side chain with ring edge) and unfavorable (negatively charged side chain with ring face) anion–aromatic interaction.

**Concluding remarks**. In summary, many previous studies have shown that protein–ligand and protein–protein complexes can be maintained in the gas phase[8–12]. Here we have shown using DMC analysis that the interaction energies of specific residue pairs can also remain unchanged upon transfer to the gas phase. The native

MS version of DMC analysis described here provides values of coupling energies without the need for multiple measurements of affinity constants and accurate determination of protein concentrations, thereby simplifying the measurements and reducing the experimental error. The method can be extended to work in a high-throughput fashion by mixing wild type and $n$ different single mutants of one protein with wild type and $m$ different single mutants of a second interacting protein. These $n + 1$ and $m + 1$ variants of the two proteins form $m \times n$ different mutant cycles that can yield $m \times n$ coupling energies that can serve as constraints for docking. Hence, it might be possible in the future to determine the structure of a complex from a single native mass spectrum if the structures of the free proteins are known.

## Methods

**Molecular biology**. The plasmid pET21d containing the wild-type genes of E9 DNase and Im9 with a His tag or E2 DNase and Im2 with a His tag was used as a template for mutagenesis by restriction-free cloning[17]. The following mutagenic

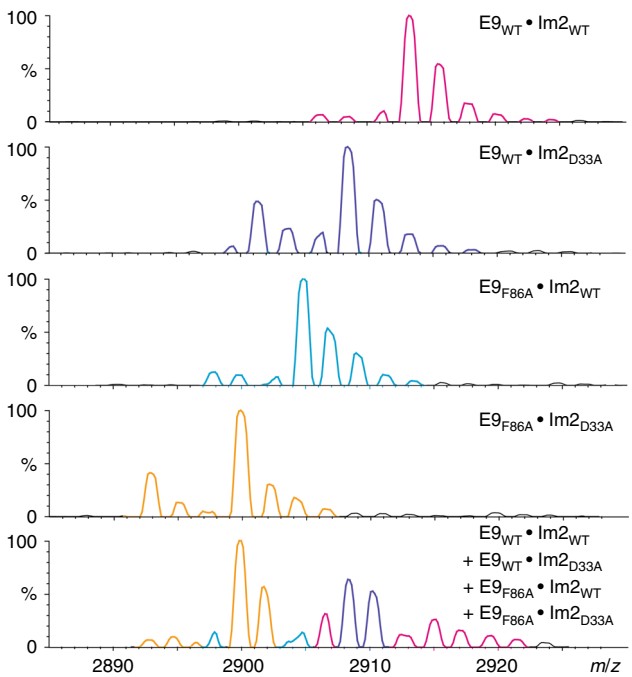

**Fig. 4** Interaction of Phe86 in E9 with Asp33 in Im2 studied by native MS. Representative mass spectra of the +9 charge state are shown for the four pairs of complexes: E9$_{WT}$ • Im2$_{WT}$ (*magenta*), E9$_{WT}$ • Im2$_{D33A}$ (*purple*), E9$_{F86A}$ • Im2$_{WT}$ (*cyan*), and E9$_{F86A}$ • Im2$_{D33A}$ (*yellow*). All these complexes were formed using equal concentrations of the two binding partners. The *spectrum in the bottom panel* corresponds to all four proteins mixed together at relative respective concentrations of E9$_{WT}$, E9$_{F86A}$, Im2$_{WT}$, and Im2$_{D33A}$ of 1:1:4:1, respectively. The highly resolved mass spectrum generated for the mixture of E9$_{WT}$, E9$_{F86A}$, Im2$_{WT}$, and Im2$_{D33A}$ enabled us to distinguish between the four different complexes that are formed

oligonucleotides were used together with the T7 universal primer as forward and backward primers, respectively.

Phe86→Ala in E9:
5′-GTGTTTCAAAAGGTTATTCTCCGGCTACTCCAAAGAA TCAACAGGTCG-3′;

Asn34→Ala in Im2:
5′-GGTGCTACTGAAGAGGATGACGCTAAATTAGTGAGAGAG TTTGAGCG-3′;

Asp33→Ala in Im2:
5′-GGTGCTACTGAAGAGGATGCCAATAAATTAGTGAGAG AGTTTGAGC-3′. The mutation Ser84→Ala in E9 was introduced together with two other mutations, His131→Ala and Arg132→Ala, designed to increase the difference between the mass of this mutant and those of the other variants without affecting its binding to Im2. This triple mutant was created using the following respective mutagenic oligonucleotides as forward and backward primers.

Ser84→Ala:
5′-GGTGCTACTGAAGAGGATGCCAATAAATTAGTGAGAGAG TTTGAGC-3′;

His131→Ala and Arg132→Ala:
5′-CCTAAGCGACATATCGATATTGCCGCAGGTAAGTAAAATGGA ACTGAAGC-3′. The mutations were confirmed by DNA sequencing of the entire genes.

**Expression and purification of the E9 and Im2 proteins**. *E. coli* BL21 (DE3) cells harboring the pET21d plasmid that contains the appropriate genes coding for E9 and Im9 with a His$_6$ tag or E2 and Im2 with a His$_6$ tag were grown in LB medium containing 100 µg ml$^{-1}$ ampicillin at 37 °C until an O.D.$_{600\ nm}$ of 0.6 was reached and protein expression was then induced by adding 1 mM isopropylthio-β-galactoside. The cells were then grown for another 4 h at 37 °C and harvested. Cell pellets were stored at −80 °C until further use. Purification was carried out by resuspending the cells in 20 mM Tris buffer (pH 7.5) containing 100 mM NaCl, 5 mM imidazole, and 1 mM DTT (buffer A) to which was added the cOmplete, EDTA-free, protease inhibitor cocktail (Roche Applied Science, Penzberg, Germany). The cells were then disrupted by sonication and the lysate was clarified by centrifugation. The supernatant was loaded on a 5 ml HisTrap HP column (Amersham Pharmacia, Uppsala, Sweden). The E9 DNase was dissociated from the column-bound Im9 and eluted by washing the column with buffer A containing 6 M guanidine hydrochloride. Im2 was eluted using a 10–200 mM imidazole gradient in buffer A containing 6 M guanidine hydrochloride. Fractions that contain the E9 and Im2 proteins were dialyzed overnight at 4 °C against water and lyophilized. The purified E9 was stored at −80 °C until further use. The Im2 protein was dissolved in 50 mM Tris buffer (pH 7.5) containing 50 mM KCl and 1 mM DTT (buffer B) and then further purified using a Superdex 75 column equilibrated with this buffer. The fractions containing Im2 were then loaded on a 5 ml HiTrap Q FF column (Amersham Pharmacia, Uppsala, Sweden). The Im2 protein was eluted from the Q FF column using a 0.05–0.5 M KCl gradient in buffer B. Fractions containing Im2 were combined, dialyzed overnight at 4 °C against water, and then lyophilized. The purified Im2 protein was stored at −80 °C until further use. The concentrations of E9 and Im2 proteins were determined using molar absorbance coefficients of 15,470 and 10,095 M$^{-1}$ cm$^{-1}$, respectively.

**Isothermal titration calorimetry**. Experiments were carried out using an ITC200 Microcal calorimeter. Lyophilized protein samples were dissolved in 250 mM ethylenediammonium diacetate (EDDA) buffer (pH 7.0) and then dialyzed overnight at 4 °C against this buffer. The protein samples were centrifuged and their concentrations determined before starting the ITC experiments. The titrations were carried out using a concentration of 0.04–1 mM E9 in the cell and a ten-fold higher concentration of Im2 in the syringe. Each ITC experiment comprised 26 injections with 3 min intervals of 1.5 µl Im2 at 25 °C. The heats of dilution were subtracted from the raw data and values of the binding constants were then determined by fitting the resulting binding isotherm to a 1:1 binding model using the ORIGIN program supplied by the manufacturer.

**Mass spectrometry analysis**. All the native MS experiments were performed using a Q Exactive Plus Orbitrap mass spectrometer (Thermo Fisher Scientific,

**Table 3 Values of the coupling energy of Ser84 in E9 with Asn34 in Im2**

| Ratio | 1:1:1:1 | | 1:2:2:1 | | 1:1:2:1 | |
|---|---|---|---|---|---|---|
| Charge state | +8 | +9 | +8 | +9 | +8 | +9 |
| $\Delta\Delta G_{int}(ij)$ (kcal mol$^{-1}$) | −0.03 ± 0.02 | −0.04 ± 0.04 | 0.01 ± 0.02 | −0.02 ± 0.04 | −0.07 ± 0.04 | −0.08 ± 0.05 |

Values of the coupling energy, $\Delta\Delta G_{int}(ij)$, were calculated for the charge states +8 and +9 using the concentration ratios 1:1:1:1, 1:2:2:1, and 1:1:2:1 as defined in Fig. 3. Similar coupling energies are obtained for the different concentration ratios

**Table 4 Values of the coupling energy of Phe86 in E9 with Asp33 in Im2**

| Ratio | 1:2:8:2 | | 1:1:4:1 | | 2:1:4:1 | |
|---|---|---|---|---|---|---|
| | +8 | +9 | +8 | +9 | +8 | +9 |
| Charge state | | | | | | |
| $\Delta\Delta G_{int}$(ij) (kcal mol$^{-1}$) | 1.08 ± 0.10 | 1.08 ± 0.05 | 1.28 ± 0.15 | 1.27 ± 0.20 | 1.32 ± 0.16 | 1.21 ± 0.07 |

Values of the coupling energy, $\Delta\Delta G_{int}$(ij), were calculated for the charge states +8 and +9 using the concentration ratios 1:2:8:2, 1:1:4:1, and 2:1:4:1 as defined in Fig. 4. Similar coupling energies are obtained for the different concentration ratios

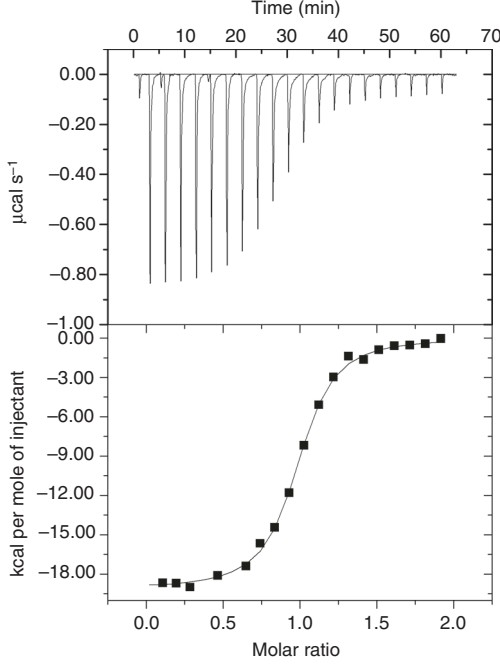

**Fig. 5** Isothermal calorimetric titration of E9 endonuclease by an Im2 mutant. Shown is a representative experiment in which wild-type E9 endonuclease was titrated with the Asn34→Ala mutant of the Im2 immunity protein. The titration was carried out in 250 mM ethylenediammonium diacetate buffer (pH 7.0) at 25 °C as described under "Methods" section

Bremen, Germany) modified for optimal transmission and detection of high molecular weight ions. Before MS analysis, proteins were dialyzed against 0.25 M EDDA buffer (pH 7). Proteins forming the DMC of interest were then mixed at the desired ratios and incubated for 5 min at room temperature. Typically, an aliquot of 2 μl protein solution was loaded into a gold-coated nano-ESI capillary prepared in-house as previously described[18] and then sprayed into the instrument. Conditions within the mass spectrometer were adjusted to preserve non-covalent interactions, with the source operating in positive mode. The following experimental parameters were used: capillary voltage, 1.7 kV; HCD direct voltage, 30–70 eV; and inlet capillary temperature in the range of 160–245 °C. MS spectra were recorded at a resolution of 17,500 (at $m/z$ 200).

**Assignment of peaks and area calculations**. For each mass spectrum, the areas of the peaks corresponding to the four possible complexes, and taking into consideration the adherence of ethylenediammonium, i.e., $(M+H)^+$ and $(M+C_2H_{10}N_2)^+$, were calculated using the PeakFit v4.12 software (Jandel Scientific, San Rafael, CA). In this program, peaks are fitted automatically to a series of Gaussians using a deconvolution approach. A numerical fitting procedure was repeated to minimize the deviation from experimental data as monitored by the coefficient of determination, $R^2$. Typical $R^2$ values were higher than 0.99. This analysis was repeated for the two detected charge states, i.e., $8^+$ and $9^+$. The reliability of the data deconvolution results was validated by comparison of the expected peak position calculated from the mass to charge ratio with the generated peakfit value. In addition, in cases of uncertainty due to the presence of bound ethylenediammonium adducts, the assignment was further confirmed by comparing the relative abundance of a specific peak in measurements performed at different ratios. For example, the peak at 2889 $m/z$ can correspond to the complex of E9$_{S84A}$ with either wild-type Im2 or Im2$_{N34A}$. This peak increases when the

concentration of wild-type Im2 is doubled relative to the other three proteins (see Supplementary Fig. 3). Hence, it was possible to assign the peak at 2889 $m/z$ to the complex of E9$_{S84A}$ with wild-type Im2. At least four independent experiments were performed for each ratio.

**Data availability**. Data supporting the findings of this study are available within the article (and its Supplementary Information file) and from the corresponding authors on reasonable request.

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

## Acknowledgements

A.H. is an incumbent of the Carl and Dorothy Bennett Professorial Chair in Biochemistry. M.S. is grateful for the support of a Starting Grant from the European Research Council (ERC) (Horizon 2020)/ERC Grant Agreement no. 636752. M.S. is an incumbent of the Aharon and Ephraim Katzir Memorial Professorial Chair.

## Author contributions

M.Sh. and A.H. designed the experiments and wrote the paper. M.So. and I.K. purified the proteins and performed the ITC measurements. J.C. and D.H. carried out the mass spectrometry experiments.

## Additional information

**Competing interests:** The authors declare no competing financial interests.

