## [Peer Review File · Nature Communications]

Reviewers' comments:

Reviewer #1 (Remarks to the Author):

The idea to use native mass spectrometry to determine protein interaction energies in the double mutant cycle methodology is a very good one and has indeed advantages over conventional types of measurements. Using the example of colicin E9 endonuclease and bacterial immunity protein Im2, the authors make a very convincing case that this concept indeed works - the agreement with standard ICT measurements is very good.

I recommend publication in Nature Commun., however subject to improvements that mainly concern clarity of presentation. Special emphasis should be put on point #5.

1. I find it strange to call the mutated proteins X₀ and Y₀, and the wild type X₁ and Y₁. I think the nomenclature should be the other way round throughout.

2. What is the meaning of the "subscript 2" in the expression $\Delta^2 G_m$? Is it just symbolizing pairwise? In this case, leave it away, because it could be mistaken for some strange mathematical operator, or for a square.

3. The authors explicitly spell out the assumption that all concerned species must have equal ionization efficiencies, and the data in Fig. 2 confirms this assumption for the mixture of the mutated and WT species, as correctly stated on Fig 6 (middle). That is not the point, though. The authors should say somewhere that the ionization efficiencies of the mixed complexes have to be equal. Is there any way to assess this?

4. The "-" signs between the mutated and unmutated version of the proteins in Fig. 2 (near the top) symbolize mixtures, not complexes. In line with other figures, simply use "+" instead. Complexes (for example in Figs. 3 and 4) could be symbolized by the fat • sign, which is normally used.

5. It is very unclear how the contributions of the different complexes are deconvoluted if there is a lot of overlap in the mass spectra, e.g. around m/z 2880 - 2890 (yellow and green species in Fig. 3) or m/z 2905 - 2915 (green, blue and red species in Fig. 4). This needs to be much better explained. Give the details in the supporting information. Likewise, it is unclear why some peaks in these mixtures are color coded to belong to one species (e.g. the two peaks in blue around m/z 2907...2910 in Fig. 3) while other peaks belonging to the same species are absent (e.g. the most intense peak in the blue spectrum in Fig. 3). Explain this in detail somewhere.

Reviewer #2 (Remarks to the Author):

Review of Double mutant cycles in the gas phase: measuring inter-protein pairwise interaction energies from a single native mass spectrum" by Sokolovski and co-workers.

This work proposes the use of native mass spectrometry together with double mutant cycles as a strategy for studying the energetic coupling between amino acids at the interface of a protein-protein complex. The demonstration and conclusions drawn are from a single, well-characterized interaction between colicin E9 endonuclease with the bacterial immunity protein Im2 with which it forms a 1:1 complex. The manuscript appears to provide data in general agreement with traditional methods based on calorimetry and in some cases in agreement with the literature. However, several issues detailed below present concerns regarding this publication.

1) Conclusions that this is a general method for interprotein interaction energy determination are based on only a single interaction, namely E9 with Im2. Previous early work with native mass

spectrometry similarly proposed to provide rapid screening of protein binding with chemical libraries (Gao et al., J. Med. Chem., 1996) showed success with some interactions but appeared to be dependent upon the type of interactions (i.e., hydrophobic vs charge-driven) and thus, gas phase stability of noncovalent interactions did not always agree with those measured in solution. Additional results would be needed with other complexes to provide a compelling case that the presented approach has broad utility. In the absence, this work would appear more restricted to the well-studied E9-Im2 complex and thus, of narrower scope.

2) On page 6, the authors indicate that additional mutations were included at His 131 and Arg 132 in E9, both mutated to Ala to allow mass difference large enough to resolve E Ser84 to Ala combined with Im2 Asn34 to Ala. The authors further claim that His 131 and Arg 132 are in the flexible tail and not involved in any intra- or inter-protein interactions which may be true based on the structures from Ref 14 (2WPT and 1EMV). However, equating flexible tail and unresolved residues in crystal structures with being unimportant for intra-molecular interactions is not universally true. Moreover, the effect on the overall E9 structure imposed by replacing these two charged residues with alanine is unknown but would likely limit flexibility in this region and may have allosteric effects elsewhere. A proper control for these additional mutations would be to measure binding energies with wild type protein and comparison with the H131, R132 mutations to demonstrate no unanticipated effects on binding energies. The authors should include these data to convince readers that these mutations are inconsequential.

3) Further discussion/clarification on the additional peaks that occur with the Im2D33A mutant-containing and Im2N34A complexes that appear to be 60 to 70 Daltons less than expected. In the E9WT-Im2N34A case (fig 3, Blue trace) these unexplained ions appear to constitute the large majority of the complex. Are these other, unanticipated mutations? How does their presence affect the concentration determination of each mutant or the overall binding energy calculation or conclusions?

4) Table 1. Errors associated with E9 wt-Im2 D33A seem large compared to others and the average K_a value differs by two orders of magnitude from the literature and should be better explained.

5) Page 22 indicates the spectra were recorded with a resolving power of 17,500. Since resolving power is a function of m/z without indicating what m/z this refers to the definition is meaningless. In addition, this raises the question of why such poor resolving power was used with a high resolution instrument. If indeed this resolving power is for $m/z=200$, then one might expect resolving power in the 2000-3000 range (where the data were recorded) to be on the order of 1000-2000. Some discussion on why such limited performance was used, especially in light of the use of direct sample infusion where acquisition of time-domain signals is not a limitation. This limitation especially curious, given the need to resolve mass differences in complexes that prompted incorporation of secondary mutations as discussed in point 2 above.

Comments of Reviewer 1

Comment 1: I find it strange to call the mutated proteins Xo and Yo, and the wild type Xi and Yi. I think the nomenclature should be the other way round throughout.

Response: Two distinct residues (i and j) are mutated in double-mutant cycle analysis. Hence, designating both of them by the same notation '0' would be confusing. Moreover, the notation '0' is appropriate for the mutated proteins since both mutations are to alanine. This notation has also been used before in the literature. Therefore, unlike the suggestions in the Comments below, we believe that accepting this one would cause confusion.

Comment 2: 'What is the meaning of the "subscript 2" in the expression Δ^2G_m ? Is it just symbolizing pairwise? In this case, leave it away, because it could be mistaken for some strange mathematical operator, or for a square.'

Response: We agree with the Reviewer that this notation, although commonly used, is potentially confusing. We've, therefore, replaced throughout the paper Δ^2G_{int} with $\Delta\Delta G_{int}$ to make clear that this term represents a double difference in free energies.

Comment 3: The authors explicitly spell out the assumption that all concerned species must have equal ionization efficiencies, and the data in Fig. 2 confirms this assumption for the mixture of the mutated and WT species, as correctly stated on Fig 6 (middle). That is not the point, though. The authors should say somewhere that the ionization efficiencies of the mixed complexes have to be equal. Is there any way to assess this?

Response: We agree with the Reviewer that it is crucial that the ionization efficiencies of the different complexes are equal and that evidence for this was not provided in the original version of the paper. We realized that a way to assess this is to compare the intensities of given concentrations of wild-type and mutant variants of one of the proteins, say E9, in the presence of an excess of a binding partner, say wild-type Im2. Under such conditions, the wild-type and mutant variants of E9 will be fully complexed with Im2 and the ratio of intensities of the complexes should correspond to the ratio of their concentrations if the ionization efficiencies of the complexes are the same. A new Supplementary Table 1 showing that this is indeed the case for three different concentrations of the various complexes has been added to the revised version.

Comment 4: The " - " signs between the mutated and unmutated version of the proteins in Fig. 2 (near the top) symbolize mixtures, not complexes. In line with other figures, simply use " + " instead. Complexes (for example in Figs. 3 and 4) could be symbolized by the fat • sign, which is normally used.

Response: We agree with the Reviewer and have replaced the "-" signs with "+" signs in Figure 2 as suggested. We have also introduced the '•' sign to indicate complexes in Figures 3 and 4.

Comment 5: It is very unclear how the contributions of the different complexes are deconvoluted if there is a lot of overlap in the mass spectra, e.g. around m/z 2880 - 2890 (yellow and green species in Fig. 3) or m/z 2905 - 2915 (green, blue and red species in Fig. 4). This needs to be much better explained. Give the details in the supporting information. Likewise, it is unclear why some peaks in these mixtures are color coded to belong to one species (e.g. the two peaks in blue around m/z 2907...2910 in Fig. 3) while other peaks belonging to the same species are absent (e.g. the most intense peak in the blue spectrum in Fig. 3). Explain this in detail somewhere.

Response: Unambiguous assignment of overlapping peaks was possible by monitoring peak intensity as a function of changing the concentration of one of the proteins. Following the Reviewer's comment, we've added a Supplementary Figure 3 that illustrates this strategy. In addition, we would like to thank the Reviewer for pointing out that peaks that appear when only one complex can form can be absent when all four complexes are formed. In the revised manuscript, we now indicate that the mixture spectra in Figures 3 and 4 were taken at 1:2:2:1 and 1:1:1:4 ratios, respectively. Therefore, some complexes will have higher tendency to form than the others. A sentence regarding this issue was added to the Legend of Fig. 3.

Comments of Reviewer 2

Comment 1: Conclusions that this is a general method for interprotein interaction energy determination are based on only a single interaction, namely E9 with Im2. Previous early work with native mass spectrometry similarly proposed to provide rapid screening of protein binding with chemical libraries (Gao et al., J. Med. Chem., 1996) showed success with some interactions but appeared to be dependent upon the type of interactions (i.e., hydrophobic vs charge-driven) and thus, gas phase stability of noncovalent interactions did not always agree with those measured in solution. Additional results would be needed with

other complexes to provide a compelling case that the presented approach has broad utility. In the absence, this work would appear more restricted to the well-studied E9-Im2 complex and thus, of narrower scope.

Response: The main advantage of the double-mutant cycle method is that it provides estimates of pairwise interaction energies that tend to be context-independent, e.g. the value it provides for a salt-bridge with a particular geometry in one protein will be similar for the same type of salt-bridge in another protein (see refs. 1-4 in the paper). This has been well established for many types of interactions and in this regard, therefore, there is no need for experiments on additional systems. Clearly, however, finding a mass spectrometry-compatible buffer might be difficult for some proteins but most methods have certain limitations. It should also be pointed out that setting-up an additional experimental system and repeating the ITC and MS experiments could take a year. We, therefore, believe that adding such data to the manuscript is beyond the scope of the current work.

Comment 2: On page 6, the authors indicate that additional mutations were included at His 131 and Arg 132 in E9, both mutated to Ala to allow mass difference large enough to resolve E Ser84 to Ala combined with Im2 Asn34 to Ala. The authors further claim that His 131 and Arg 132 are in the flexible tail and not involved in any intra- or inter-protein interactions which may be true based on the structures from Ref 14 (2WPT and 1EMV). However, equating flexible tail and unresolved residues in crystal structures with being unimportant for intra-molecular interactions is not universally true. Moreover, the effect on the overall E9 structure imposed by replacing these two charged residues with alanine is unknown but would likely limit flexibility in this region and may have allosteric effects elsewhere. A proper control for these additional mutations would be to measure binding energies with wild type protein and comparison with the H131, R132 mutations to demonstrate no unanticipated effects on binding energies. The authors should include these data to convince readers that these mutations are inconsequential.

Response: We agree with the Reviewer that showing experimentally that the tail mutations have no effect on binding of E9 S84A to Im2 is preferable to assuming this to be the case because of structural considerations. We, therefore generated the E9 S84A single mutant (without the two tail mutations) and measured its binding to Im2 using ITC. The results show that the tail mutations do indeed have no effect on binding. The data are shown in a new Supplementary Figure (Supplementary Figure 1 in the revised text) and mentioned in the text.

Comment 3: Further discussion/clarification on the additional peaks that occur with the Im2D33A mutant-containing and Im2N34A complexes that appear to be 60 to 70 Daltons less than expected. In the E9WT-Im2N34A case (fig 3, Blue trace) these unexplained ions appear to constitute the large majority of the complex. Are these other, unanticipated mutations? How does their presence affect the concentration determination of each mutant or the overall binding energy calculation or conclusions?

Response: The additional peaks are not due to unanticipated mutations or modifications since only a single peak with a mass corresponding to the calculated mass was observed upon analysis of each of the proteins under denaturing conditions (Supplementary Figure 2). These peaks are most likely due to bound ethylenediammonium ions, which add 60 Da

to the calculated protein mass. Their presence did not affect concentration determinations or any of the energy calculations since we were able to assign them to specific complexes (see response to Comment 5 of Reviewer 1).

Comment 4: Table 1. Errors associated with E9 wt-Im2 D33A seem large compared to others and the average K_a value differs by two orders of magnitude from the literature and should be better explained.

Response: We obtained a value of $\sim 0.25 \mu\text{M}^{-1}$ for the association constant of wild-type E9 to the Im2 D33A mutant when using 50 mM MOPS buffer (pH 7.0) containing 200 mM NaCl as in ref. 13. An error for the association constant for this interaction was not reported in ref. 13 probably because it was relatively large owing to tight binding. The discrepancies between the values reported here and the literature values are, therefore, most likely due to the different buffers and salt concentrations used in the two studies. A comment regarding this was added as a footnote to Table 1 as suggested by the Reviewer.

Comment 5: Page 22 indicates the spectra were recorded with a resolving power of 17,500. Since resolving power is a function of m/z without indicating what m/z this refers to the definition is meaningless. In addition, this raises the question of why such poor resolving power was used with a high resolution instrument. If indeed this resolving power is for $m/z=200$, then one might expect resolving power in the 2000-3000 range (where the data were recorded) to be on the order of 1000-2000. Some discussion on why such limited performance was used, especially in light of the use of direct sample infusion where acquisition of time-domain signals is not a limitation. This limitation especially curious, given the need to resolve mass differences in complexes that prompted incorporation of secondary mutations as discussed in point 2 above.

Response: We thank the Reviewer for raising this point, and have corrected the text accordingly to indicate that MS spectra were recorded at a resolution of 17,500 at m/z 200. In general, we performed a systematic analysis in order to define the optimal resolving power for our mass measurements. While the increase in resolving power improved the resolution it reduced the signal-to-noise. In the case of this specific system, a resolving power of 17,500 was found to yield the optimal spectra.

In summary, we believe that the review process has improved the work. Thank you for your consideration.

REVIEWERS' COMMENTS:

Reviewer #1 (Remarks to the Author):

The authors have addressed the comments & questions in a satisfactory way.

I would, however, strongly suggest to not use the subscript "0" for the mutated protein. "0" is associated with something like "baseline" or "unmodified". How about denoting the modified protein with subscript "m"? That would make the nomenclature much more clear.